# Expression of FGFR1–4 in Malignant Pleural Mesothelioma Tissue and Corresponding Cell Lines and its Relationship to Patient Survival and FGFR Inhibitor Sensitivity

**DOI:** 10.3390/cells8091091

**Published:** 2019-09-16

**Authors:** Gregor Vlacic, Mir A. Hoda, Thomas Klikovits, Katharina Sinn, Elisabeth Gschwandtner, Katja Mohorcic, Karin Schelch, Christine Pirker, Barbara Peter-Vörösmarty, Jelena Brankovic, Balazs Dome, Viktoria Laszlo, Tanja Cufer, Ales Rozman, Walter Klepetko, Bettina Grasl-Kraupp, Balazs Hegedus, Walter Berger, Izidor Kern, Michael Grusch

**Affiliations:** 1University Clinic for Respiratory and Allergic Diseases Golnik, 4204 Golnik, Slovenia; gregor.vlacic@klinika-golnik.si (G.V.); katja.mohorcic@klinika-golnik.si (K.M.); tanja.cufer@klinika-golnik.si (T.C.); ales.rozman@klinika-golnik.si (A.R.); izidor.kern@klinika-golnik.si (I.K.); 2Translational Thoracic Oncology Laboratory, Division of Thoracic Surgery, Department of Surgery, Medical University of Vienna, 1090 Vienna, Austria; mir.hoda@meduniwien.ac.at (M.A.H.); thomas.klikovits@meduniwien.ac.at (T.K.); katharina.sinn@meduniwien.ac.at (K.S.); elisabeth.gschwandtner@meduniwien.ac.at (E.G.); balazs.dome@meduniwien.ac.at (B.D.); viktoria.laszlo@meduniwien.ac.at (V.L.); walter.klepetko@meduniwien.ac.at (W.K.); 3Institute of Cancer Research, Department of Medicine I, Medical University of Vienna, 1090 Vienna, Austria; karin.schelch@meduniwien.ac.at (K.S.); christine.pirker@meduniwien.ac.at (C.P.); barbara.peter-voeroesmarty@meduniwien.ac.at (B.P.-V.); jelena.brankovic@meduniwien.ac.at (J.B.); bettina.grasl-kraupp@meduniwien.ac.at (B.G.-K.); walter.berger@meduniwien.ac.at (W.B.); 4Department of Tumor Biology, National Koranyi Institute of Pulmonology, 1085 Budapest, Hungary; 5Department of Thoracic Surgery, National Institute of Oncology-Semmelweis University, 1085 Budapest, Hungary; 6Department of Thoracic Surgery, University Medicine Essen-Ruhrlandklinik, 45239 Essen, Germany; balazs.hegedues@rlk.uk-essen.de

**Keywords:** malignant pleural mesothelioma, FGFR, overall survival, immunohistochemistry, infigratinib sensitivity

## Abstract

Malignant pleural mesothelioma (MPM) is a devastating malignancy with limited therapeutic options. Fibroblast growth factor receptors (FGFR) and their ligands were shown to contribute to MPM aggressiveness and it was suggested that subgroups of MPM patients could benefit from FGFR-targeted inhibitors. In the current investigation, we determined the expression of all four FGFRs (FGFR1–FGFR4) by immunohistochemistry in tissue samples from 94 MPM patients. From 13 of these patients, we were able to establish stable cell lines, which were subjected to FGFR1–4 staining, transcript analysis by quantitative RT-PCR, and treatment with the FGFR inhibitor infigratinib. While FGFR1 and FGFR2 were widely expressed in MPM tissue and cell lines, FGFR3 and FGFR4 showed more restricted expression. FGFR1 and FGFR2 showed no correlation with clinicopathologic data or patient survival, but presence of FGFR3 in 42% and of FGFR4 in 7% of patients correlated with shorter overall survival. Immunostaining in cell lines was more homogenous than in the corresponding tissue samples. Neither transcript nor protein expression of FGFR1–4 correlated with response to infigratinib treatment in MPM cell lines. We conclude that FGFR3 and FGFR4, but not FGFR1 or FGFR2, have prognostic significance in MPM and that FGFR expression is not sufficient to predict FGFR inhibitor response in MPM cell lines.

## 1. Introduction

Malignant pleural mesothelioma (MPM) is a devastating malignancy arising from mesothelial cells lining the chest cavity. Asbestos is the main causative agent for MPM but the latency period between exposure and MPM manifestation can be more than 40 years [1]. While strict regulations on the use of asbestos have been implemented in many countries, there is still widespread use and mining of asbestos in parts of the world leading to an ongoing rise in global incidence [2]. MPM is highly refractory to conventional therapies and the prognosis is generally poor with a median overall survival of little more than one year. Chemotherapy with cisplatin and pemetrexed yields a modest survival benefit, which can be slightly improved only in selected patients by addition of bevacizumab and combination with surgery and/or radiotherapy as additional treatment modalities [3]. Despite multiple clinical studies investigating targeted therapies in MPM, no effective new treatments have been identified in this area, while immunotherapy seems to be moderately effective in a subgroup of patients [3,4]. Genomic analysis of MPM has identified recurrent mutations and structural aberrations mostly in tumor suppressor genes including BAP1, NF2, TP53, SETD2, and CDKN2A, which are difficult to target directly [5,6]. However, there is also compelling evidence for hyperactivation of growth- and survival-promoting signals in several pathways including the Hippo [7,8], phosphatidylinositol 3-kinase (PI3K) [9], and fibroblast growth factor receptor (FGFR) [10,11] signaling axes, that could provide more ‘druggable’ targets. Others and we have previously reported the overexpression of FGFR1 and several FGFs in MPM cell models and tissue specimens [10,11,12]. Moreover, we have described the growth-promoting and EMT-inducing capabilities of FGF2 in MPM cells, identified the miR-16 family members as regulators of FGFR1 and FGFR4 [13] and characterized in preclinical models the potential benefit of combining FGFR inhibition with chemotherapy or radiation [11,14]. Recently, a link between FGFR inhibitor sensitivity, FGF9/18 mediated FGFR3 activation, and loss of BAP1 was established [15]. Nevertheless, a comparative analysis of the expression of all four FGFRs in MPM tissues has not been performed so far.

In the current investigation, we therefore focus on expression of the four existing FGFRs (FGFR1–4) in MPM tissue and corresponding patient-derived cell lines as well as their relationship to MPM prognosis and potential prediction of response to FGFR kinase inhibition.

## 2. Materials and Methods

### 2.1. Clinical Samples

Patients: 94 MPM patients were included in the study and full clinical follow_up data was available in 81 patients, 41 from Austria (Medical University of Vienna, Vienna, Austria) and 40 from Slovenia (University Clinic for Respiratory and Allergic Diseases Golnik, Golnik, Slovenia). All patients were referred for diagnosis and treatment to one of the two institutions between 2006 and 2015. MPM diagnosis was histologically confirmed during routine clinical work-up in all patients. Patients were staged clinically and pathologically according to the IMIG staging system [16]. Details on patients’ characteristics and treatment modalities are depicted in Table 1.

Tumor samples: All tumor samples were obtained during diagnostic procedures or at the time of surgery (macroscopic complete resection). Histological specimens were fixed in formalin and embedded in paraffin (FFPE). One 3 µm section from a representative, tumor-rich FFPE block was stained by hematoxylin/eosin to confirm and locate malignant areas and consecutive sections were used for FGFR1–4 immunohistochemistry. Clinical data and tumor blocks were retrospectively collected for all cases according to the corresponding local ethic committees (Ethical Committee of University of Vienna; Ethical approval number: 904/2009; Date: 9 December 2019).

### 2.2. Immunohistochemistry

Primary antibodies against FGFR1 (sc-121-G), FGFR2 (sc-122), FGFR3 (sc-123), and FGFR4 (sc-124) from Santa Cruz Biotechnology (Dallas, TX, USA) have been extensively used and characterized in multiple tissues and cell models [17,18,19] and were used here as previously described [20]. In brief, 3 µm sections were cut from FFPE blocks. Immunohistochemistry was performed on a Ventana Benchmark XT (Ventana Medical Systems, Tucson, AZ, USA) for antibodies FGFR2–4. Pretreatment with CC1 for 56 min (CC1: Ventana 950-124) was performed, incubation time for primary antibody was 32 min at 37 °C (for FGFR4 60 min), dilution 1:50, counterstaining with hematoxylin (Ventana: 760-2021) and bluing reagent (760-2037) was done for 4 min each. For FGFR1, incubation with primary antibody was performed at a 1:50 dilution for 1 h at ambient temperature and subsequent processing was done with the ImmPRESS HRP Anti goat IgG polymer detection kit (Vector Laboratories, Peterborough, United Kingdom) as per the manufacturer’s recommendations. A semi-quantitative three tier scoring system was used to assess the intensity of the cytoplasmic and/or nuclear staining—0 if there was no staining, 1 if there was faint positivity in at least 50% of tumor cells, and 2 if there was an unequivocally strong positivity in at least 50% of tumor cells.

### 2.3. Cell Lines

Establishment of cell lines from clinical samples followed previously described protocols [21]. Cells were cultured in RPMI medium with 10% fetal bovine serum (FBS) at 37 °C in a humidified atmosphere with 5% CO_2_. Cell line authentication was done by array comparative genomic hybridization and STR analysis [11,22]. All cell lines were regularly checked for mycoplasma contamination. 

### 2.4. Generation of Cell Blocks

Cells were grown to about 80% confluency in 10 cm dishes, washed with cold PBS and scraped into 15 mL tubes in PBS. After centrifugation, the supernatants were discarded and the pellets were fixed in 3 mL 10% formaldehyde for 2 h at room temperature. Afterwards, cells were centrifuged again and excess formaldehyde was discarded. HistoGel (HG-4000-12, Thermo Fisher Scientific, Waltham, MA, USA) was liquefied in boiling water for 5 min and cell pellets were resuspended in equal volumes (ca. 50 µL) of HistoGel. The HistoGel cell mixture was pipetted as a single drop per sample onto a parafilm mounted on a cold plate. After 30 min, the solidified drops were placed into 3 mL of 70% ethanol and kept at 4 °C until paraffin embedding. Further processing and immunostaining were performed using the same antibodies and procedures as for the tissue specimens.

### 2.5. Drug Treatment

For determining FGFR inhibitor sensitivity, 3000 cells per well were seeded into 96-well plates in growth medium containing 10% FCS. After 24 h for recovery, cells were treated with concentrations ranging from 0.1 to 10 µM of the FGFR1–3 inhibitor infigratinib (BGJ398) or the FGFR4-specific inhibitor BLU9931. Controls received vehicle (DMSO) only. After 72 h, DNA content of cells was analyzed by SYBR green detection as previously published [13]. IC_50_ values were determined from dose–response curves with GraphPad Prism (GraphPad Software, San Diego, CA, USA). 

### 2.6. Quantitative Real-Time Reverse Transcription PCR (qRT-PCR)

Cells were grown in 25 cm^2^ flasks to about 80% confluence and total RNA was extracted with the innuPREP RNA mini kit (Analytik Jena, Jena, Germany) according to the manufacturer´s instructions. RNA was reverse transcribed with M-MLV reverse transcriptase (Thermo Fisher Scientific) and the resulting cDNAs were used as templates for qRT-PCR analysis with Taqman assays (FGFR1: Hs00913142m1, FGFR2: Hs01552918m1, FGFR3: Hs00179829m1, FGFR4: Hs01106908m1, all from Thermo Fisher Scientific). Relative transcript levels were calculated as 2^−ΔC^t × 10^5^ normalized to the two house-keeping genes GAPDH (Hs99999905m1) and beta-actin (Hs99999903m1). 

### 2.7. Statistical Analysis

Categorical data was compared by performing Fishers’ exact or chi-square tests. Overall survival (OS) was defined as time between initial MPM diagnosis and date of death or last follow-up. OS was estimated by the Kaplan–Meier method and a log rank test was used to calculate survival differences between two groups. Pearson´s correlation coefficients were determined to explore the relationship between two continuous variables. All results were considered statistically significant when P < 0.05 two-sided. Analyses were performed using the SPSS Statistics 23.0 package (SPSS Inc., Chicago, IL, USA) and GraphPad Prism.

## 3. Results

### 3.1. FGFR1 and FGFR2 Show Strong Staining in MPM Tissue

Staining for FGFR1–4 was performed in tumor tissue from 94 MPM patients and evaluated using a three-tier semi quantitative scoring system. Representative examples of staining are shown in Figure 1A. Overall, the tumors showed a strong staining for FGFR1 and FGFR2, whereas FGFR3 and FGFR4 showed weaker and more restricted expression (Figure 1B). The staining was mostly granular and evenly distributed for FGFR1 and FGFR2, localized in the cytoplasm and in the nucleus with a higher intensity in the latter. The cases that stained positive for FGFR3 and FGFR4 showed a more scattered distribution of staining with focal areas of more intensive staining and other areas with faint but still convincingly positive staining. A less intense—mostly cytoplasmic—reaction was occasionally seen in some areas that appeared to be composed mostly of non-tumoral cells. 

### 3.2. Expression of FGFR3 and FGFR4 But Not of Other FGFRs Correlates with Patient Survival

Clinical follow_up data was available for 81 patients. All patients showed a strong FGFR2 staining and therefore no correlation with clinical parameters was performed in this group. Staining for FGFR1 and FGFR3 was not correlated with age, sex, histology, stage, or type of treatment (Appendix A), whereas FGFR4 staining was found in 4 of 22 females but only 2 of 59 males (Appendix A). Due to the small numbers of biphasic (N = 13) and sarcomatoid (N = 7) tumors, these two categories were combined as non-epithelioid for statistical analyses. Despite the absence of significant correlations between staining pattern and histology for any of the FGFRs, an overall trend towards low or absent staining was observed for sarcomatoid tumors (Figure 2A, Appendix A).

FGFR1 staining intensity did not show a significant correlation with patient survival and also the four patients with complete absence of FGFR1 staining had similar OS (Figure 2B). Only 2 patients had a staining score of 2 for FGFR3 and therefore were pooled with the 32 patients with a staining score of 1. Forty-seven patients showed absence of FGFR3 staining and had a significantly longer OS (P = 0.043). FGFR4 expression, was detectable only in 6 of 81 patients (7.4%) and showed a highly significant correlation with a shorter OS (P = 0.0027). 

### 3.3. FGFR Inhibitor Sensitivity in Patient-Derived Cell Lines

From 13 patients analyzed in our study, establishment of stable cell lines was successful. These cell lines were tested for their response to the FGFR inhibitor infigratinib. Cell viability was determined after 72 h and IC_50_ values were calculated from dose–response curves (Figure 3A, Table 2). Of the 13 cell lines, 3 had an IC_50_ value below 0.5 µM and were classified as sensitive, 4 cell lines had calculated IC_50_ levels above 10 µM or showed no inhibition even at the highest concentration and were classified as resistant. The remaining six cell lines showed intermediate sensitivity. When sensitive, resistant and intermediate MPM cell lines were compared with respect to FGFR staining scores of the corresponding tissue specimens, there was no correlation between FGFR inhibitor sensitivity and the score of either FGFR1 or FGFR3 (Figure 3B, FGFR2 is not shown due to indiscriminately high staining in all samples). A similar picture also emerged when the sum of the scores for all FGFRs (not shown) or for all FGFRs with the exception of FGFR4 (which is poorly inhibited by infigratinib) was considered.

### 3.4. FGFR Expression in Patient-Derived Cell Lines

Since the staining pattern of the tumors showed considerable heterogeneity and it was unclear whether cell lines derived from a tumor would have the same FGFR expression as the corresponding tumor, we generated cell blocks from all 13 cell lines of our panel and stained them according to the same protocol that had been used for the tumor specimens. Interestingly, the staining patterns of the cell lines were much more homogenous than those of the tumors they had been derived from (Figure 4A and Table 2). All 13 cell lines showed strong staining for FGFR1 and FGFR2, weak staining for FGFR3 and lacked detectable levels of FGFR4 (Table 2, Figure 4A). Due to this very homogenous protein expression pattern, it was obvious that the observed differences in sensitivity to infigratinib cannot be explained by expression differences in FGFR receptor proteins. 

Therefore, we determined mRNA levels of FGFR1–4 in the whole cell line panel to be able to compare mRNA and protein levels and see whether FGFR mRNA levels might be better predictors of FGFR inhibitor sensitivity of the cell lines than staining patterns. The highest mRNA levels were found for FGFR1, whereas FGFR2, 3, and 4 were expressed at considerably lower levels (Figure 4B). Indeed, even the lowest mRNA expression level of FGFR1 was several fold higher than the highest level of any other FGFR. Plotting FGFR1 expression levels against infigratinib sensitivity showed no correlation between FGFR1 mRNA level and infigratinib IC_50_ (Figure 4C, R^2^ = 0.041, P = 0.5485). Despite their lower expression levels, we also compared FGFR2–4 mRNA expression to infigratinib sensitivity, but no significant correlations were observed (Appendix A). There was, however, a positive correlation between FGFR3 and FGFR4 mRNA expression and a weak inverse correlation between FGFR1 and FGFR2 mRNA expression (Appendix A). 

Finally, since FGFR4 was the only FGFR linked to shorter survival in MPM patients and since FGFR4 is poorly inhibited by infigratinib, we also tested the FGFR4-specific inhibitor BLU9931 in our cell line panel. Overall, IC_50_ values of BLU9931 were about an order of magnitude higher than those of infigratinib, and three of the cell lines showed no inhibition (Appendix A). While the correlation between high FGFR4 mRNA and BLU9931 sensitivity did not reach statistical significance, notably, the two cell lines with the highest FGFR4 mRNA showed the highest sensitivity towards the FGFR4 inhibitor (Figure 4D).

## 4. Discussion

Tyrosine kinase inhibitors (TKI) targeting mainly FGFRs (like infigratinib [23] or erdafitinib [24]) or co-targeting FGFRs in addition to VEGFR and other RTKs (like nintedanib [25] or lenvatinib [26,27]) are considered as promising therapeutic agents for a number of different malignancies. Nevertheless, it has remained largely unclear which patients have the highest likelihood of achieving a benefit from these agents. Gene amplifications, mutations, and translocations have been reported for FGFRs in various malignancies. FGFR1 amplification, for instance, occurs in non-small cell lung cancer (NSCLC) and breast cancer patients [28]. Fusion oncogenes containing parts of FGFR1–3 and a number of different partners were found in smaller percentages of patients with glioblastoma, bladder cancer, and a number of additional malignancies [29]. Point mutations of FGFR4 were reported in rhabdomyosarcoma [30]. While some of these genetic FGFR alterations were shown to predict for sensitivity to FGFR inhibitor treatment [23], in other studies (e.g., in lung and breast cancer) the correlation between FGFR1 amplification and FGFR inhibitor sensitivity was less clear [31,32]. In addition to structural alterations in FGFRs, also abberant expression patterns of the four FGFRs are being explored for associations with disease course and response to chemotherapy and kinase inhibitors, for instance in glioblastoma, gastric cancer, and skin cancers [33,34,35]. Moreover, FGFRs are also expressed on stromal cells and were shown to mediate interaction of cancer cells and cancer-associated fibroblasts [36].

In MPM, several reports have excluded genetic aberrations in specific FGFRs to occur at significant frequencies but have reported sensitivity of subgroups of MPM cells to FGFR inhibition by pharmacologic and genetic approaches [10,11,15]. Results from our group show that strongly reduced ERK and AKT phosphorylation in response to treatment with the FGFR inhibitor PD166866 occurs only in sensitive MPM cells [11]. This suggests that repressing FGFR-dependent hyperactivation of both pathways is one prerequisite for FGFR inhibitor sensitivity. In line with this, it was shown that antiproliferative effects of the multikinase inhibitor sorafenib were most pronounced in tumor-initiating mesothelioma cells which expressed high levels of FGF2, leading to autocrine activation of FGFR1 [37]. These cells showed a repression of high basal ERK and MEK phosphorylation levels by sorafenib in the absence of detectable basal AKT phosphorylation. Marek et al. found a link between high FGFR1 protein expression determined by immunoblotting in MPM cell lines and sensitivity to the FGFR inhibitors ponatinib and FP1039/GSK3052230 [10]. The latter inhibitor was recently tested in a phase Ib study in MPM [38]. Quispel-Janssen et al. reported that loss of BAP1 was linked to increased FGFR inhibitor sensitivity and elevated FGFR/FGF expression in MPM cells [15]. The multikinase inhibitor nintedanib targeting VEGFR, FGFR, and PDGFR has shown promising results in preclinical [22] and early clinical trials in MPM [39] but failed to show a benefit in a subsequent phase III study [40]. However, since the clinical study was intended to test nintedanib as an antiangiogenic therapy, no biomarkers were used in this study to select patients on the basis of tumor cell expression of specific nintedanib targets.

Our data confirm that co-expression of several FGFRs is common in MPM tissue. This is in line for instance with hepatocellular carcinoma (HCC) [20] where recent data suggest high expression of FGFR2 and FGFR3 as potential biomarkers for infigratinib response [41]. Our data, however, did not reveal a connection between FGFR expression and infigratinib efficacy. The partly nuclear staining of FGFRs in tumor cells was previously noted by others and us and has been linked to invasive behavior for instance in breast cancer [11,42]. In contrast to the widely expressed FGFR1 and FGFR2, FGFR3 and FGFR4 staining identified subgroups of MPM patients with significantly worse prognosis. FGFR4 was previously reported as a growth driver in conjunction with FGF19 overexpression in HCC [43,44,45] and has been shown to be of negative prognostic impact in NSCLC [46]. Interestingly, FGF1, FGF2, and FGF18, which are the three most abundantly expressed FGFs in MPM cells [11], all have high affinity for FGFR4 [47]. High FGF2 levels in serum and pleural effusions were previously shown to correlate with poor patient survival in MPM [48], whereas for other FGFs data are largely missing. Strategies for targeting FGFR4 are currently being developed [49,50], and, while further research is required, it seems encouraging in this respect that the MPM cell lines with the highest FGFR4 mRNA expression show the strongest inhibition by the FGFR4-specific agent BLU9931. 

FGFR3 was previously shown to be relevant for sensitivity to the FGFR inhibitor AZD4547 in MPM cells by transmitting FGF9 signals [15]. Moreover, increased expression of FGFR3 was shown in that study to be associated with mutated BAP1 in gene expression data of MPM patients. In our data, FGFR3 expression was generally weaker and more restricted on protein and mRNA level when compared to FGFR1 and correlated with shorter OS in MPM patients but not with infigratinib sensitivity in cell lines. Of the three sensitive cell lines in our panel, two had a deletion or mutation in BAP1, whereas one showed no obvious alteration. Mutated BAP1, however, was also present in one of the three resistant cell lines [51]. 

An obvious discrepancy exists with respect to the strong FGFR2 staining and comparatively low mRNA expression data in our cell lines. This could either be explained by cross-reactivity of the antibody with additional antigens or by the presence of FGFR2 splice variants not detected by the Taqman probe used for mRNA analysis. A notable observation of our study is the much more homogenous staining in cell lines compared to the corresponding tumors. This could indicate that culture conditions could favor specific expression patterns either via selection for the survival of cells with, e.g., high FGFR1 expression or via providing a more homogenous microenvironment that leads to a more homogenous expression. 

Overall, our study demonstrates that FGFR3 and FGFR4 have prognostic value in MPM, whereas the other FGFRs are often co-expressed but do not correlate with patient survival or clinicopathologic parameters. Further studies in larger patient collectives will be required to clarify the prognostic value of FGFR expression for the different histological subtypes of MPM. Our data confirm the existence of an FGFR inhibitor sensitive subgroup of MPM cell lines, but we have not found a satisfactory predictive capacity to identify sensitive cell lines in our panel by analyzing FGFR1–4 expression. This leads us to postulate that additional, still undiscovered factors play a major part in controlling FGFR inhibitor sensitivity in MPM.

## Figures and Tables

**Figure 1 cells-08-01091-f001:**
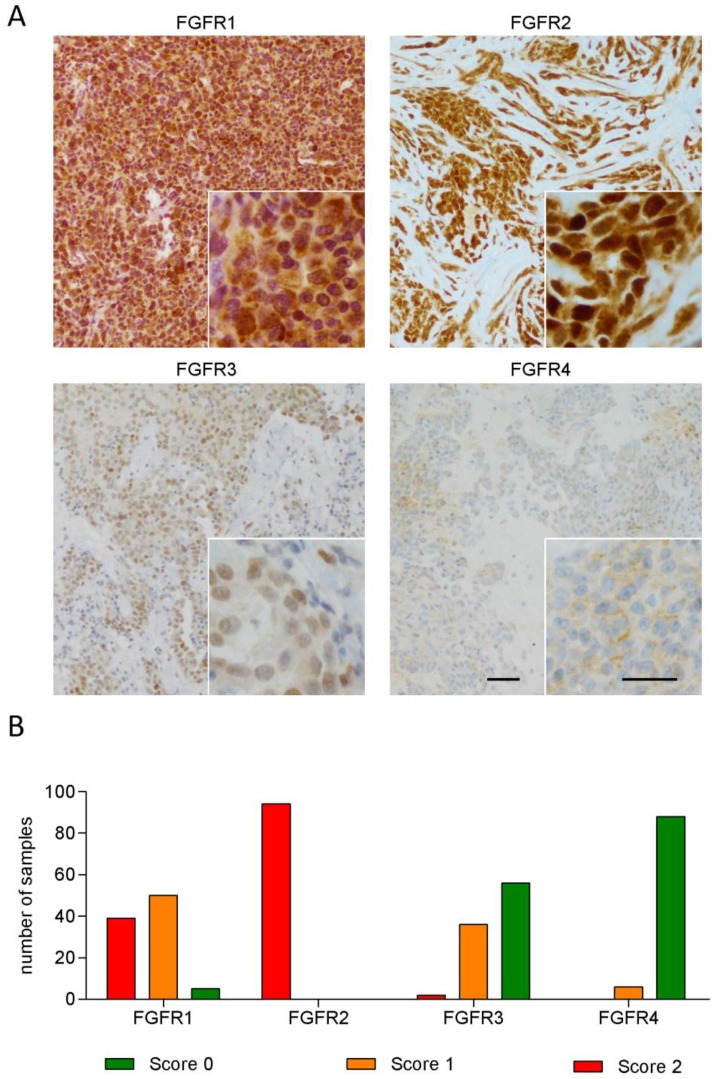
Expression of FGFR1–4 in MPM tissue. (**A**) Representative images of MPM tissue specimens stained for FGFR1 (score 2, epithelioid), FGFR2 (score 2, epithelioid), FGFR3 (score 1, epithelioid), and FGFR4 (score 1, epithelioid). Scale bar = 25 µm. (**B**) Distribution of staining intensities of FGFR1–4 in 94 MPM tissue specimens.

**Figure 2 cells-08-01091-f002:**
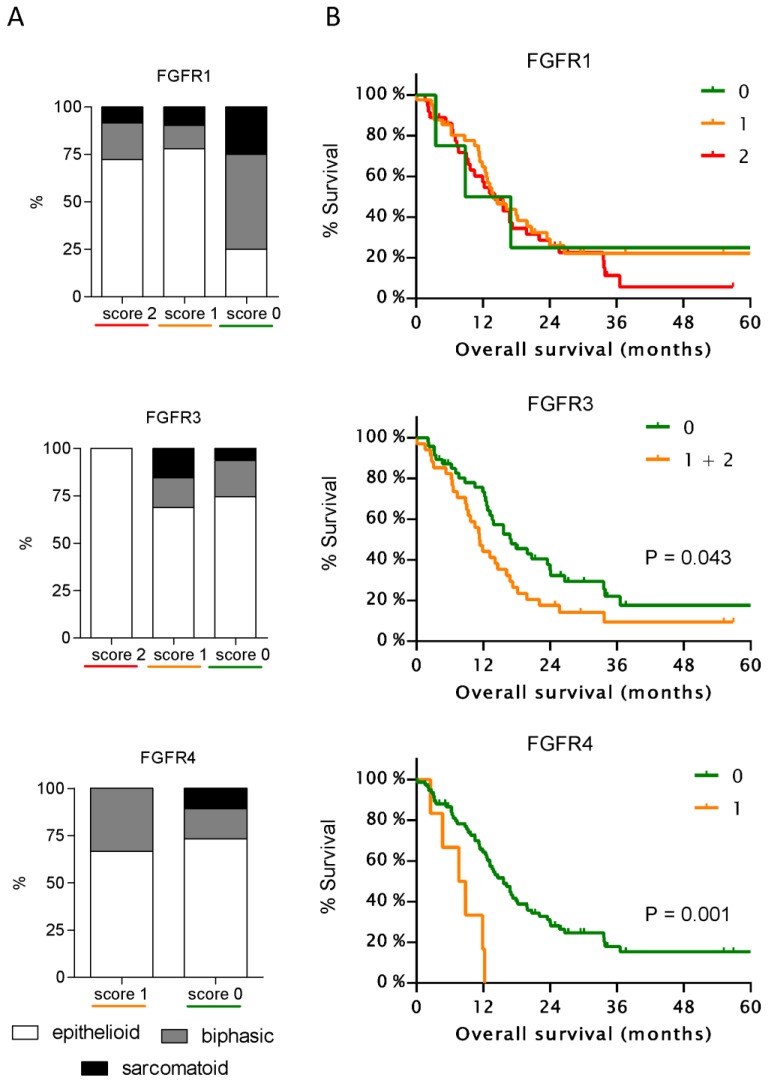
Correlation of FGFR expression with histology and patient prognosis. (**A**) Percentage of epithelioid, biphasic, and sarcomatoid tumors within the different staining groups of FGFR1 (upper panel), FGFR3 (middle panel), and FGFR4 (lower panel). (**B**) Kaplan–Meier curves for overall survival of MPM patients with different staining scores of FGFR1 (upper panels), FGFR3 (middle panel), and FGFR4 (lower panel).

**Figure 3 cells-08-01091-f003:**
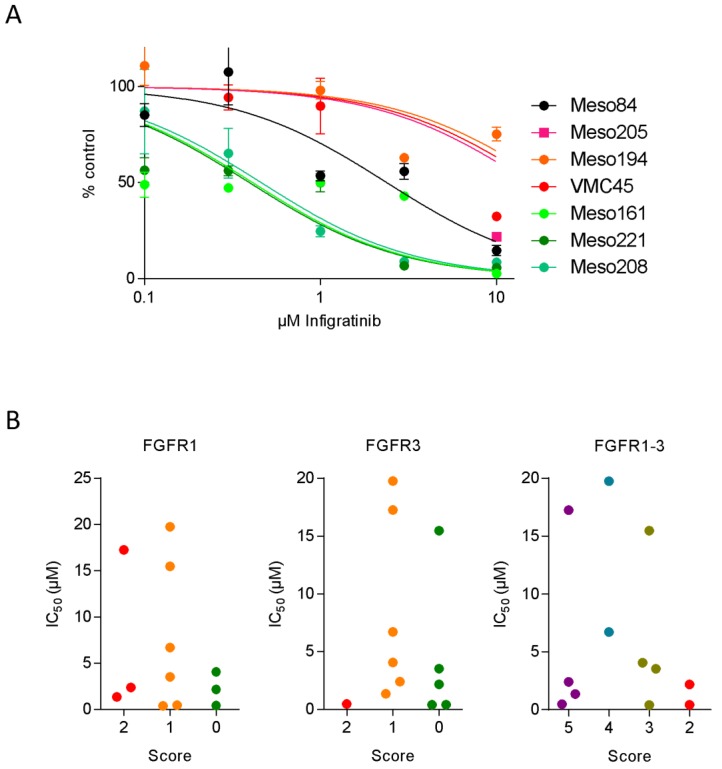
Sensitivity of patient-derived cell lines to the FGFR inhibitor infigratinib. (**A**) MPM cell lines were incubated with increasing concentrations of infigratinib or vehicle (DMSO) as control and cell number was determined after 72 h. Dose–response curves were calculated with GraphPad Prism. Three sensitive cell lines (IC_50_ < 1 µM), one intermediate cell line (1 µM < IC_50_ < 10 µM), and three resistant cell lines (IC_50_ > 10 µM) are shown in green, black, and red, respectively. (**B**) Infigratinib IC_50_ values of the cell lines were plotted against the IHC scores for FGFR1, FGFR3, or the sum of FGFR1–3 of the corresponding tumors from which the cell lines were established.

**Figure 4 cells-08-01091-f004:**
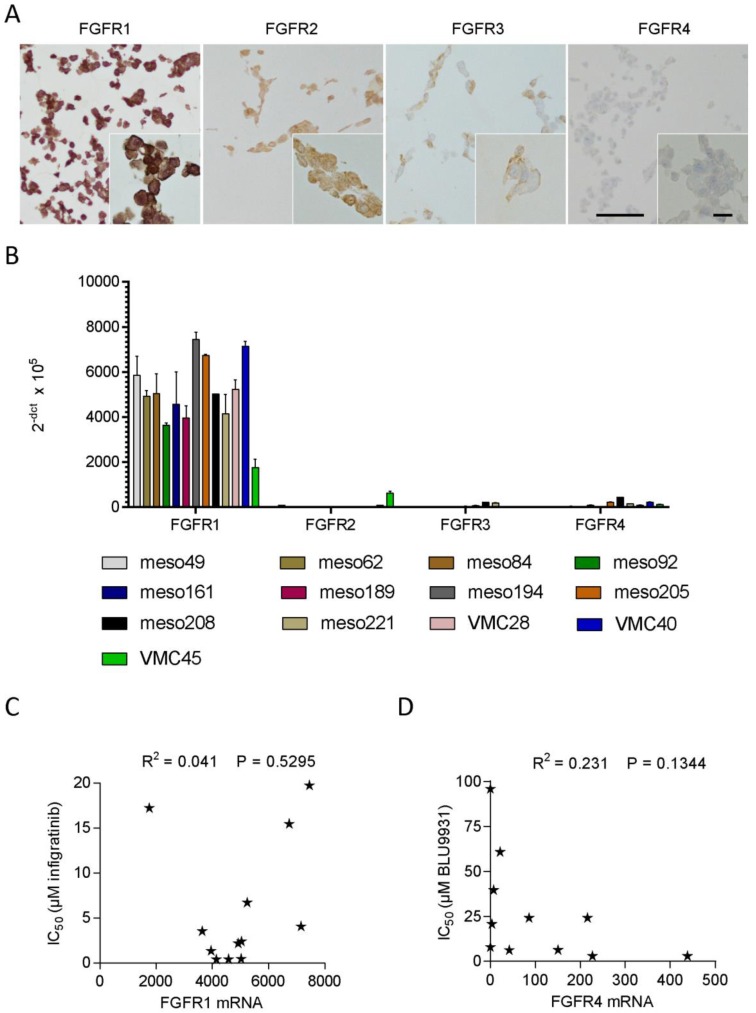
Expression of FGFR1–4 in patient-derived cell lines. (**A**) Representative images from cell blocks stained for FGFR1 (Meso208), FGFR2 (Meso161), FGFR3 (VMC45), and FGFR4 (Meso205). Scale bar = 100 µm in overview images and 10 µm in high magnification insets. (**B**) Total mRNA was isolated from logarithmically growing cell lines and subjected to cDNA synthesis. Quantitative RT-PCR was performed by Taqman assays for FGFR1–4. Expression levels were plotted as 2^−dCt^ × 10^5^ normalized to the house-keeping genes GAPDH and beta-actin. (**C**) Infigratinib IC_50_ values were plotted as function of FGFR1 mRNA expression level. (**D**) BLU9931 IC_50_ values were plotted as function of FGFR4 mRNA expression levels.

**Table 1 cells-08-01091-t001:** Patient characteristics

All Patients		*n*	%
Age	<60	22	27
	≥60	59	73
Sex	female	22	27
	male	59	73
Histology	non-epithelioid	22	27
	epithelioid	59	73
Stage	early	28	35
	late	53	65
Treatment overview	BSC	15	19
	CHT	33	41
	CHT + RT	3	4
	CHT + S	11	14
	TMT	19	23

BSC: best supportive care; CHT: chemotherapy; RT: radiotherapy; S: surgery; TMT: trimodality therapy.

**Table 2 cells-08-01091-t002:** Cell line characteristics

Cell Line	Histology	IC_50_ (µM)	Cell Block IHC Score			
		Infigratinib	FGFR1	FGFR2	FGFR3	FGFR4
Meso49	bi	n.i.	2	2	1	0
Meso62	sarc	2.18	2	2	1	0
Meso84	sarc	2.39	2	2	1	0
Meso92	bi	3.53	2	2	1	0
Meso161	bi	0.41	2	2	1	0
Meso189	epi	1.34	2	2	1	0
Meso194	epi	19.76	2	2	1	0
Meso205	epi	15.47	2	2	1	0
Meso208	epi	0.46	2	2	1	0
Meso221	epi	0.39	2	2	1	0
VMC28	epi	6.71	2	2	1	0
VMC40	bi	4.06	2	2	1	0
VMC45	epi	17.24	2	2	1	0

epi: epithelioid; bi: biphasic; sarc: sarcomatoid; n.i.: no inhibition.

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
