# Peer review of "Expression of FGFR1–4 in Malignant Pleural Mesothelioma Tissue and Corresponding Cell Lines and its Relationship to Patient Survival and FGFR Inhibitor Sensitivity"

_cells, 2019, doi:10.3390/cells8091091_

Round 1

Reviewer 1 Report

The authors have submitted a MS where they provide evidence that FGFR3 and FGFR4 are prognostic in MPM, whereas other FGFRs do not correlate with patient survival  . Whereas cells were sensitive to FGFr signaling inhibition the authors have found no satisfactory predictive capacity  sensitive  by analyzing FGFR1‐4 expression and suggest other factors could play a role to determine the sensitivity to this inhibition

The MS is clear a nd the results consistent although more informations will be reinforcing what proposes

a) these two papers should be quoted and properly commented

https://www.ncbi.nlm.nih.gov/pubmed/?term=Mutti+L.+FGF

b) AKT and ERK signaling before and after FGFr inhibition should be shown or discussed as for literature already published on this matter

Author Response

We thank the reviewer for the encouraging comments on our manuscript.

In response to comment a) of the reviewer we now reference and discuss the following literature:

The inhibition of FGF receptor 1 activity mediates sorafenib antiproliferative effects in human malignant pleural mesothelioma tumor-initiating cells. Pattarozzi A, Carra E, Favoni RE, Würth R, Marubbi D, Filiberti RA, Mutti L, Florio T, Barbieri F, Daga A. Stem Cell Res Ther. 2017 May 25;8(1):119. doi: 10.1186/s13287-017-0573-7.

(lines 292-296, reference 33)

Basic fibroblast growth factor in mesothelioma pleural effusions: correlation with patient survival and angiogenesis. Strizzi L, Vianale G, Catalano A, Muraro R, Mutti L, Procopio A. Int J Oncol. 2001 May;18(5):1093-8.

(lines 316-317, reference 43)

We previously published data regarding the impact of FGFR inhibition on AKT and ERK signals (Fibroblast growth factor receptor inhibition is active against mesothelioma and synergizes with radio- and chemotherapy. Schelch K, Hoda MA, et al. Am J Respir Crit Care Med. 2014 Oct 1;190(7):763-72. doi: 10.1164/rccm.201404-0658OC.). In response to comment b) of the reviewer we now discuss these data as well as data from Marubbi et al regarding ERK and AKT signaling following FGFR inhibition in the discussion section of the manuscript (lines 289-296).

Reviewer 2 Report

My main concern about this manuscript is the antibodies used for the immunohistochemistry. The majority of the conclusions of the study are based on the performance of these antibodies. Yet, there are some discrepancies between the mRNA and IHC results in the cell lines, and the staining in tumours is mainly cytoplasmatic and nuclear, which is unusual for transmembrane receptors. The authors need to detail what steps were taken to demonstrate specificity of staining and include corresponding pictures (such as control tissues or cell lines that are known to express/not express these proteins; and/or negative staining using a blocking peptide; and/or a single clear band of expected size on western blotting performed with the cell lines extracts). A part from this technical issue, the manuscript is interesting, well written and original and it investigates an important issue. 

Author Response

We thank the reviewer for the encouraging comments on our manuscript and for highlighting the importance of antibody performance. Indeed, all antibodies used in our study have been extensively characterized and used for IHC and IF as well as immunoblotting in different tissues and disease models in multiple previous publications for instance in New England Journal of Medicine (Sanna-Cherchi et al, N Engl J Med 2013), Nature Genetics (Shingu et al, Nat Genet 2017), Journal of Thoracic Oncology (Donnem et al, JTO 2009), PLoS Genetics (Borad et al, PLoS Genetics 2014) or PLOS Biology (Bogani et al, PLoS Biology 2009). Some of the previous investigations have demonstrated the specificity of these antibodies by using blocking peptides (Saucedo et al, PLoS One 2015 for FGFR1-3, Streit et al, Br J Cancer 2006 for FGFR4), whereas our group has demonstrated positive staining of FGFR1-4 in liver cancer tissue known to express these receptors (Gauglhofer et al, Hepatology 2011). In the methods section (lines 105-107, references 17-19) of our revised manuscript we include references to some previous papers characterizing antibody performance. Others and we have also previously performed immunoblots with FGFR1-4 antibodies. However, since FGFRs are expressed in multiple splice isoforms and are subject to posttranslational modifications and signaling-induced processing and endocytosis followed by degradation, immunoblots typically show several FGFR-specific bands. These signaling-induced effects may also be responsible for the previously described partially cytoplasmic and nuclear localization of FGFRs (e.g. Chioni and Grose J Cell Biol 2012, discussed as reference 38).

Reviewer 3 Report

The authors investigated the prognosis of survival including also the response to treatment analyzed in tissue cultures of 94 cases of rare malignancy  - malignant pleural mesothelioma with very low survival rate independently of the stage of the disease. They used tissue cell lines established  from clinical samples. They determined FGFR inhibitor antitumor properties in tissue culture using FGFR1-3 inhibitor infigratinib or FGFR4-specific inhibitor BLU9931 activity. The manuscript is very interesting because the authors were able to collect the samples of very rare malignancy and were able to establish the stable cell lines from 13 patients. However, due to the number of cases included to the analysis, three different subtypes and different treatment regimens it is too difficult to draw conclusions about the prognostic value of biomarkers under study.

Author Response

We thank the reviewer for the encouraging feedback. We agree that additional studies including more samples will have to confirm the prognostic value of FGFRs in MPM and elucidate whether there are differences in prognostic impact with respect to the histological subtypes. In the revised version, this statement has now been added to the conclusions (lines 339-340). As the reviewer has pointed out, MPM is a rare malignancy and therefore collection of large sample numbers is a difficult task. We are convinced that our data provide important novel information about the role of FGFRs in MPM and will stimulate further work on this subject